# Iron-based trinuclear metal-organic nanostructures on a surface with local charge accumulation

Cornelius Krull[1], Marina Castelli[1,2], Prokop Hapala[3], Dhaneesh Kumar[1,2,4], Anton Tadich[5], Martina Capsoni[6], Mark T. Edmonds[1,2,4], Jack Hellerstedt [1,2,3], Sarah A. Burke [6,7,8], Pavel Jelinek [3,9] & Agustin Schiffrin [1,2,4]

Coordination chemistry relies on harnessing active metal sites within organic matrices. Polynuclear complexes—where organic ligands bind to several metal atoms—are relevant due to their electronic/magnetic properties and potential for functional reactivity pathways. However, their synthesis remains challenging; few geometries and configurations have been achieved. Here, we synthesise—via supramolecular chemistry on a noble metal surface—one-dimensional metal-organic nanostructures composed of terpyridine (tpy)-based molecules coordinated with well-defined polynuclear iron clusters. Combining low-temperature scanning probe microscopy and density functional theory, we demonstrate that the coordination motif consists of coplanar tpy's linked via a quasi-linear tri-iron node in a mixed (positive-) valence metal–metal bond configuration. This unusual linkage is stabilised by local accumulation of electrons between cations, ligand and surface. The latter, enabled by bottom-up on-surface synthesis, yields an electronic structure that hints at a chemically active polynuclear metal centre, paving the way for nanomaterials with novel catalytic/magnetic functionalities.

[1] School of Physics & Astronomy, Monash University, 19 Rainforest Walk, Clayton 3800, Australia. [2] Monash Centre for Atomically Thin Materials, Monash University, 20 Research Way, Clayton 3800, Australia. [3] Institute of Physics of the CAS, Cukrovarnicka 10, Prague 16200, Czech Republic. [4] ARC Centre of Excellence in Future Low-Energy Electronics Technologies, Monash University, 19 Rainforest Walk, Clayton 3800, Australia. [5] Australian Synchrotron, 800 Blackburn Road, Clayton Victoria 3168, Australia. [6] Department of Physics and Astronomy, University of British Columbia, 6224 Agricultural Road, Vancouver, British Columbia, Canada V6T 1Z1. [7] Department of Chemistry, University of British Columbia, 2036 Main Mall, Vancouver, British Columbia, Canada V6T 1Z1. [8] Stewart Blusson Quantum Matter Institute, University of British Columbia, 2355 East Mall, Vancouver, British Columbia, Canada V6T 1Z4. [9] RCPTM, Palacky University, Šlechtitelů 27, 783 71 Olomouc, Czech Republic. Correspondence and requests for materials should be addressed to P.J. (email: jelinekp@fzu.cz) or to A.S. (email: agustin.schiffrin@monash.edu)

Metal-organic molecular complexes and coordination polymers allow for a vast range of functionalities, both technological and biological, from catalysis to light harvesting to gas storage, sensing, and exchange[1–3]. In these processes, the atomic-scale electronic configuration of the metal centres plays a crucial role; it dictates the chemical reactivity of the molecular systems[1–3]. For instance, polypyridyl-based complexes, due to their coordination morphology and resulting optoelectronic and chemical properties, can be used for photovoltaics[4] and catalysis[5]. Polynuclear complexes are of special interest, since magnetic[6] and electronic interactions between metal atoms in close proximity can give rise to useful properties (e.g., cooperative electronic and steric effects), with multiple active metal centres potentially enhancing the catalytic processes in comparison to mononuclear systems[7–9]. Wet chemistry synthetic methods have achieved an array of different polynuclear complexes, exhibiting direct metal-to-metal bonds[10,11]. Based on both inorganic compounds and organic matrices stabilising transition metal centres, a variety of geometries have been obtained, e.g., trigonal nodes, linear nanochains up to eight atoms long[10]. However, the synthesis of such compounds remains challenging, with only a limited number of configurations available, and metals and ligands utilised. Notably, only a few[10] metal-to-metal compounds based on iron (Fe)—an important element for catalysis[12]—exist.

On-surface supramolecular chemistry[13]—a bottom-up approach driven by noncovalent intermolecular interactions and metal–ligand bonding on a surface—allows for the design of self-assembled, atomically precise metal-organic nanomaterials, with morphologies, properties and functionalities that can differ dramatically from those obtained via conventional synthetic chemistry. Such approaches can provide an alternative for synthesising functional complexes with polynuclear metal sites. Confining the molecular and atomic building units to two-dimensions (2D) can lead to the stabilisation of morphologies/functionalities not achievable in three-dimensions (3D) and can be beneficial for device-based applications, e.g., solid-state heterogeneous catalysis[14], where the active materials are required to interface with a solid. The surface, via its structural symmetry and chemical reactivity, can further provide a means of control over atomic-scale morphology and electronic/chemical properties. These methods have recently allowed for the synthesis of low-dimensional metal-organic systems with di-nuclear coordination metal centres (e.g., di-iron complexes with promising catalytic[15] and magnetic[16] properties). Higher-order polynuclear systems have also been achieved, e.g., tri- and tetra-copper coordination[15,17]. However, multinuclear complexes with direct metal-to-metal interactions (and their resulting properties)[8] have not been rigorously demonstrated on surfaces. A trinuclear macromolecular complex based on the coordination of a terpyridine (tpy)-containing molecule with copper adatoms from a copper surface has been proposed[17], but without a direct atomic-scale characterisation of the coordination motif.

Here, we report the bottom-up on-surface synthesis and atomic-scale characterisation of a tri-nuclear Fe complex, resulting from the one-dimensional (1D) self-assembly of a tpy-based aromatic molecule (tpy–phenyl–phenyl–tpy; TPPT) coordinated with Fe adatoms on a Ag (111) surface. Using a combination of low-temperature scanning tunnelling microscopy (STM) and spectroscopy (STS), non-contact atomic force microscopy (ncAFM), local contact potential difference (LCPD) measurements, density functional theory (DFT), and ncAFM image simulations, we directly demonstrate that the metal–ligand coordination motif consists of flat, coplanar tpys linked via a quasi-linear tri-iron cluster. The Fe atoms in this coordination node are in a cationic mixed-valence configuration, exhibiting direct metal–metal bonding. Importantly, the coordination is concomitant with a charge redistribution, with electrons accumulating at the cation–ligand surface interface, stabilising the metal-organic linkage, and potentially leading to catalytic activity localised at the multinuclear centre. Our results position on-surface supramolecular chemistry as a pathway to synthesise functional multinuclear coordination nodes from bottom-up, with configurations and properties that would not be feasible via conventional synthetic chemistry approaches without a surface.

## Results

**On-surface self-assembly of 1D metal-organic nanostructures.** Figure 1b shows an STM image of TPPT molecules on Ag(111) (see inset of Fig. 1b for chemical structure and Methods for sample preparation). As observed for other aromatic molecules on noble metals[18], TPPT adsorbs flat and is imaged as a "dogbone" feature with "v-shaped" tpys, resembling its Lewis structure (see Ref. [19] for more details). Subsequent deposition of Fe results in the formation of self-assembled 1D nanostructures (Fig. 1c). We observed metal-organic chains (MOCs) with lengths of several tens of molecular units, that is, on the order of 100 nm. These MOCs consist of TPPT molecules with their tpys arranged in a head-to-head motif, stabilised by coordination with Fe adatoms. In contrast to pristine TPPT[19], the adsorption orientation of the MOCs does not seem to strongly correlate with the substrate crystalline axes; the metal-organic chaining reduces the molecule-surface interaction (see Supplementary Fig. 12 for more details).

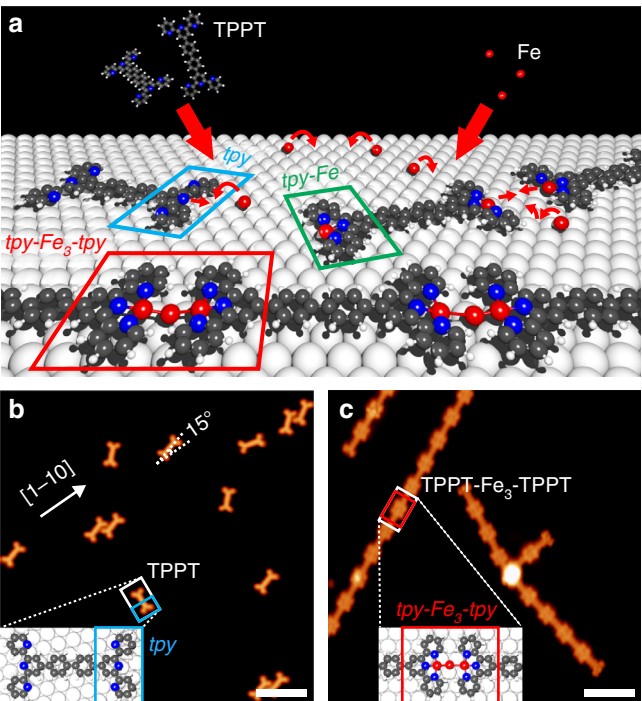

**Fig. 1** Self-assembly of 1D metal-organic nanostructures on a noble metal surface. **a** Schematic of the bottom-up synthesis of metal-organic nanochains (MOCs) via on-surface terpyridine (tpy)–iron (Fe) coordination. TPPT molecules and Fe adatoms were sequentially deposited from the gas phase onto the Ag(111) surface held at room temperature. **b** STM image of TPPT molecules adsorbed on Ag(111) ($I_t = 5$ pA, $V_b = 20$ mV). **c** STM image of MOCs resulting from the on-surface metal–ligand coordination ($I_t = 25$ pA, $V_b = 20$ mV). The STM tip was functionalised by picking up a single carbon monoxide molecule (see Methods). Insets: DFT calculated the energetically favourable adsorption geometries of TPPT (**a**) and MOC [(**b**); black: carbon; blue: nitrogen; red: iron; grey: silver]. Scale bars: 5 nm

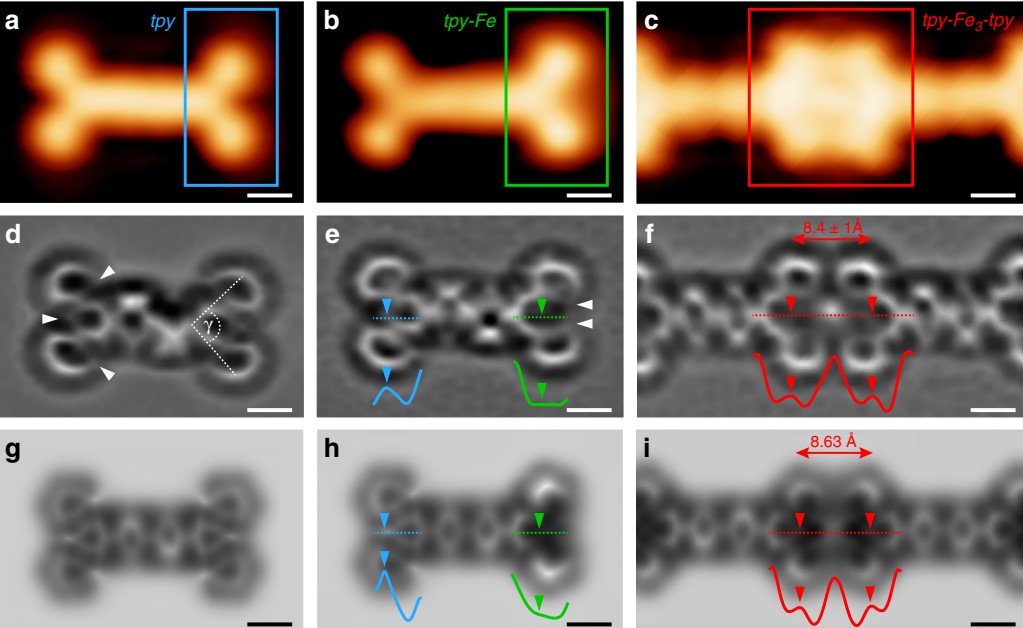

**Fig. 2** Intramolecular morphology of the TPPT–Fe coordination motif. An isolated TPPT (**a**, **d**, **g**), a singly metalated TPPT (**b**, **e**, **h**) and a metal-organic coordination node in a chain (**c**, **f**, **i**) are shown. **a–c** STM images with CO-functionalised tip ((**a**): $I_t = 5$ pA, $V_b = 20$ mV; (**b**, **c**): $I_t = 10$ pA, $V_b = 20$ mV). **d–f** Constant-height Laplace-filtered ncAFM images with CO-functionalised tip [tip height is 0.8 Å above that determined by the STM set point $I_t = 25$ pA, $V_b = 20$ mV measured on bare Ag(111)]. **g–i** Simulated ncAFM images based on DFT calculations (Methods). Coordination node in (**i**) includes three linearly arranged Fe atoms. Locations of N atoms correspond to lower intensity areas of *pyr* [indicated by white arrows in (**d**) and (**e**)]. Inset curves: apparent height profiles [arb. u.] along the dashed lines. Scale bars: 5 Å

In Figs. 2a–c, we imaged TPPT with STM in different stages of coordination: single pristine TPPT, TPPT with one of its *tpys* coordinated with an Fe adatom, and two TPPTs linked by Fe within a MOC. The non-metalated left *tpy* of the molecule in Fig. 2b is imaged identically to that of the pristine molecule (Fig. 2a), with its characteristic "v-shape". The right metalated *tpy* in Fig. 2b appears brighter with a central protrusion due to the interaction with Fe. The metal-organic linkage in the MOC consists of apparently flat *tpys* in a symmetric head-to-head configuration with a central protrusion (Fig. 2c). STM neither resolves the intramolecular conformation nor the atomic-scale configuration of the metal-organic node mediating the chaining.

**Atomic-scale morphology of the coordination motif**. To elucidate the atomic-scale morphology of 1D nanostructures, we performed ncAFM with CO-functionalised tips (see Methods). This technique allows for real-space imaging with sub-molecular resolution[20]. Figs. 2d–f show CO-tip ncAFM images of the same systems as in Figs. 2a–c. The image of pristine TPPT (Fig. 2d) is similar to the Lewis structure of the gas-phase molecule, confirming quasi-flat adsorption. The two phenyl (*ph*) rings are slightly tilted out of the molecular plane (see Supplementary Table 1 and Supplementary Fig. 6). The pyridine (*pyr*) of the *tpy* appear as distorted hexagons[21], very subtly tilted towards the surface due to interactions between nitrogen lone electron pairs and the noble metal[19,22]. NcAFM imaging of the singly metalated TPPT (Fig. 2e) shows a left non-metalated *tpy* identical to that of the pristine molecule (Fig. 2d), consistent with STM; the central *ph*s remain tilted. Imaging of the right *tpy* (coordinated with one Fe adatom) differs from the left non-metalated one. The distal *pyr*s are tilted, with the lower side oriented towards the centre of *tpy*, indicating a rotation around the C–C bond to enable coordination between the N atoms (Fig. 2d white arrows) and Fe. This coordination-induced conformational change is expected[23]. The in-plane angle γ between distal *pyr* is reduced by metalation

(Fig. 2d; Supplementary Table 1). The axial *pyr* appears darker in comparison to that of the non-metalated *tpy*, with its N-termination barely visible. This indicates an increased attraction between the CO tip and the *tpy–Fe*, and a bending of this *pyr* due to bonding between Fe and surface (Fig. 2e profiles), consistent with previous work on molecules interacting with metal adatoms[22]. The ncAFM map in Fig. 2f shows a coordination node in a MOC, with two opposing *tpys* and a central bright feature. We estimate[21] a distance of $8.4 \pm 1$ Å [(standard deviation (s.d.)] between the opposing axial N-atoms, more than twice as large as that of the analogous, single metal complex in solution[24]. Each *tpy* within the node is similar to *tpy–Fe* in Fig. 2e: the darker sides of distal *pyr*s are rotated towards the centre of *tpy*; the axial *pyr* shows the same darker depression due to increased tip-surface attractive force. We deduce that the coordination node is composed of at least two Fe atoms, with each of the two *tpys* interacting with one adatom (as for *tpy–Fe* in Figs. 2b, e). Imaging of the MOCs (including coordination nodes) is independent of their orientation with respect to the underlying substrate (Supplementary Fig. 12).

To determine the atomic-scale morphology of the metal-organic linkage, we deconstructed a Fe–TPPT node in a MOC via STM lateral manipulation (Fig. 3a, b), revealing its composition: a *tpy* imaged identically to a *tpy* metalated with a single Fe adatom (Fig. 3c), labelled as $tpy–Fe^{(D)}$ (green box), and a *tpy* imaged with an elongated protrusion (orange box, $tpy–Fe_2^{(D)}$). Analogously, by manipulating a single Fe adatom towards a *tpy* coordinated with a single Fe adatom ($tpy–Fe^{(A)}$), we assembled a *tpy* ($tpy–Fe_2^{(A)}$; Fig. 3e) imaged identically to $tpy–Fe_2^{(D)}$. This provides evidence that $tpy–Fe_2^{(D)}$ consists of a *tpy* coordinated to two Fe adatoms, similar to di-nuclear poly-*pyr* coordination chains on Au(111)[25]. We did not observe spontaneous self-assembly of this doubly metalated *tpy–Fe_2* complex.

We performed $dI/dV$ STS (Figs. 3f–i) to corroborate that the deconstructed species ($tpy–Fe^{(D)}$, $tpy–Fe_2^{(D)}$) are identical to the

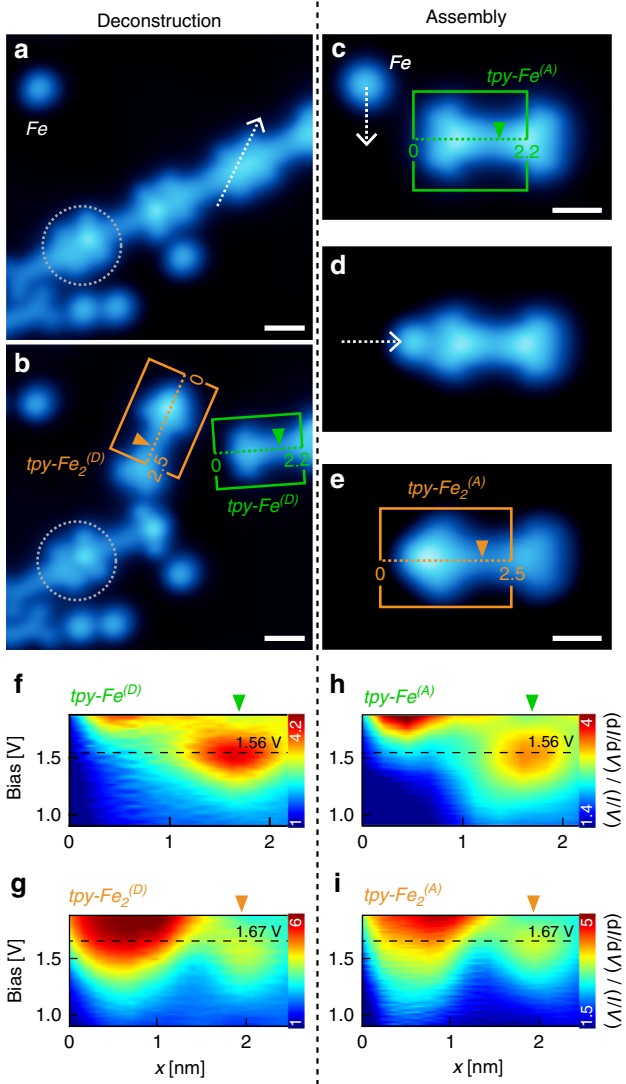

**Fig. 3** Direct demonstration of tri-iron coordination node via STM manipulation. **a**, **b** STM images ($V_b = -500$ mV; a: $I_t = 50$ pA; b: $I_t = 400$ pA) before and after lateral tip movement (white dashed arrow) across coordination node in MOC. The node is composed of two types of metalated *tpys*: *tpy*–$Fe_2^{(D)}$ and *tpy*–$Fe^{(D)}$. Grey dashed circle indicates coordination node decorated with a CO molecule (Supplementary Fig. 2). **c**–**e** STM images ($I_t = 100$ pA, $V_b = -500$ mV) of the assembly of a doubly metalated *tpy* (*tpy*–$Fe_2^{(A)}$) by manipulating a single Fe atom towards a singly metalated *tpy* (*tpy*–$Fe^{(A)}$). Imaging of singly metalated *tpy* (*tpy*–$Fe^{(D)}$ and *tpy*–$Fe^{(A)}$) is different from Fig. 2b due to tunnelling parameters and no CO tip functionalisation. **f**, **i** ($dI/dV$)/($I/V$) spectra acquired along singly and doubly metalated *tpy*'s (orange and green dashed lines in (**b**, **c**, **e**); see Supplementary Fig. 8). Singly (doubly) metalated *tpy* shows tunnelling resonance at 1.56 V (1.67 V, respectively) associated with the unoccupied state at the centre of TPPT. Set point: $I_t = 25$ pA, $V_b = -1$ V. Topographic and electronic properties of *tpy*–$Fe^{(D)}$ (*tpy*–$Fe_2^{(D)}$) are identical to those of *tpy*–$Fe^{(A)}$ (*tpy*–$Fe_2^{(A)}$, respectively). STM manipulation and imaging performed with a Ag-terminated Pt/Ir tip. Scale bars: 1 nm

respective assembled ones (*tpy*–$Fe^{(A)}$, *tpy*–$Fe_2^{(A)}$). The ($dI/dV$)/($I/V$) spectra in Fig. 3h (*tpy*–$Fe^{(A)}$) show tunnelling resonances at ∼ +1.56 V (associated with an empty orbital at the centre[19] of TPPT) and >+1.8 V (related to the metalated *tpy*). This spectroscopic signature is identical to that of *tpy*–$Fe^{(D)}$ (Fig. 3f). The spectra in Fig. 3i (*tpy*–$Fe_2^{(A)}$) show resonances at ∼ +1.67 V (centre of molecule) and >+1.5 V (metalated *tpy*), equivalent to

those of *tpy*–$Fe_2^{(D)}$ (Fig. 3g). Our topographic and spectroscopic data provide compelling evidence that the metal-organic node consists of a singly metalated *tpy*–Fe group and a doubly metalated *tpy*–$Fe_2$, that is, a cluster of three Fe atoms arranged quasi-linearly in a plane perpendicular to the surface.

We calculated the energetically favourable configurations of the molecular systems of interest via DFT (Methods). The inset in Fig. 1c shows the relaxed adsorption geometry of the tri-iron node, with each Fe atom located close to a Ag(111) hollow site. Simulated ncAFM images based on these calculations (Fig. 2g–i; Methods and Supplementary Table 2) reproduce our experiments: (i) the outwards pointing N atoms of the slightly tilted distal *pyr's* of the pristine TPPT (Fig. 2g); (ii) the coordination-mediated rotation of distal *pyr's*, with the N atoms oriented towards the centre of the dark, singly metalated *tpy*–Fe (Fig. 2h); (iii) the quasi-linear tri-iron coordination node with the central bright feature, indicating a repulsive electrostatic interaction between the CO tip and the node (see Supplementary Fig. 9). Remarkably, the experimental and simulated apparent height line profiles (curves in Fig. 2d, e, h, i) are in excellent agreement, for a range of tip-sample distances (see Supplementary Figs. 3 and 7).

**Local charge redistribution at the coordination centre.** To understand the central protrusion in the ncAFM image in Fig. 2f, we performed LCPD mapping. This method can provide insight into intramolecular charge distribution of the system[26,27]. In a MOC (Fig. 4a), the LCPD values $V^*$ show significant contrast between the TPPT ligand, Fe centre and Ag substrate. This is indicative of a negatively charged TPPT due to substrate-to-molecule electron transfer, similar to other aromatic systems[18]. Importantly, the map in Fig. 4b shows a maximum $V^*$ at the centre of the node, consistent with the accumulation of negative charge.

Figure 4e shows the LCPD along a metalated *tpy*–Fe (green curve) and a coordination node (red). The LCPD of the ligand within a MOC is larger [$11 \pm 6$ mV (s.d.)] than for the isolated metalated TPPT. This is consistent with the larger energy of the ($dI/dV$)/($I/V$) resonance associated with an empty molecular state of *tpy*–$Fe_2$ in comparison to *tpy*–Fe (Fig. 3h, i). This larger energy —indicating a smaller electron affinity for *tpy*–$Fe_2$—could be explained by a larger negative charge at the ligand due to further metalation of *tpy*. The LCPD at the Fe of the isolated metalated TPPT (green curve, Fig. 4e) does not show the increase observed at the centre of the coordination node (red curve); the latter is a signature of the tri-iron coordination and hints towards a nontrivial charge redistribution at the node, not observed for the analogous single Fe atom organometallic complex in solution[23].

To explore this, we considered the DFT-calculated charge density at the node and the resulting electron density difference (Methods); see Fig. 4f. The latter shows electron depletion around the Fe cores (blue) and electron accumulation (red) between the Fe, N, and Ag atoms, indicative of Fe–N coordination bond formation. Notably, the electrons between the Fe atoms point towards the formation of a metal–metal bond, with Fe–Fe distances [$\sim 2.4 \pm 0.2$ Å (s.d.)] comparable to other Fe-based multinuclear systems[10]. Moreover, significant negative charge from the Ag substrate contributes to stabilising the node.

The electron density involved in the Fe–Fe bonds and at the node-Ag interface (Fig. 4f) gives rise to a repulsive electrostatic interaction with the (negatively charged) CO molecule on the tip, that is, a negative electrostatic potential above the node (Fig. 4c). This is responsible for the increase of the LCPD (Fig. 4b) and for the protrusion at the centre of the tri-iron linkage seen in ncAFM imaging (Fig. 2f–i; Supplementary Fig. 9)[28]. These features are due to the node bonding motif and resulting electronic properties and electrostatic field, rather than the topography (the central Fe

lies ~0.55 Å closer to the surface than the adjacent ones; the Fe trimer is quasi-linear and slightly kinked towards the surface; Fig. 4d). Although assigning an integer oxidation state to a metal

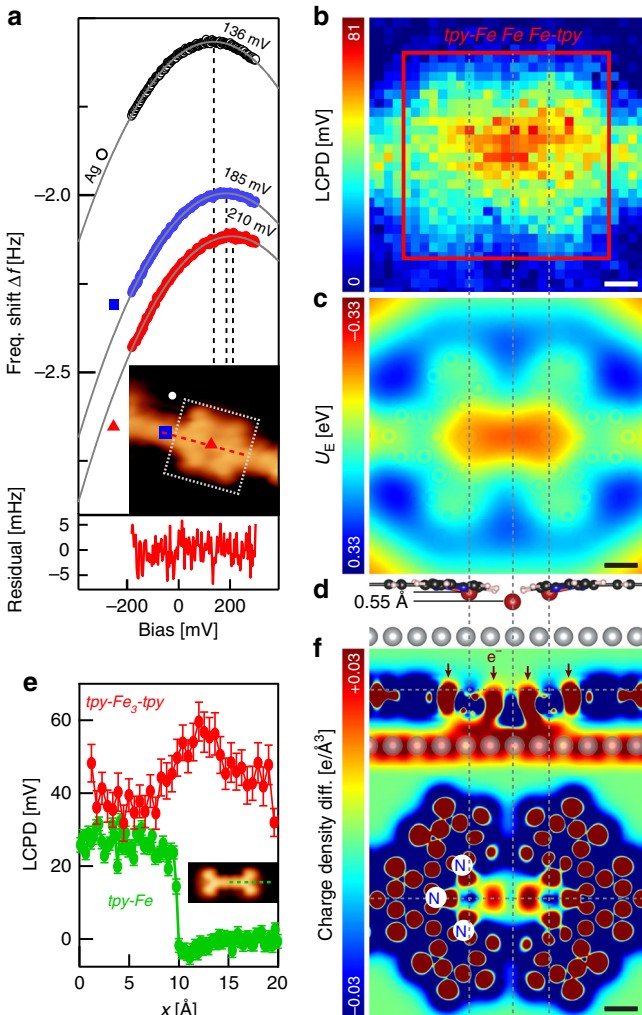

**Fig. 4** Charge redistribution and electron accumulation at the centre of the tri-iron coordination node. **a** Frequency shift $\Delta f$ as a function of sample bias voltage $V_b$, measured at the centre of the tri-iron node (red), on TPPT molecule in MOC (blue) and on bare Ag(111) (white). Data recorded at constant height, 3 Å above that defined by an STM set point ($I_t = 25$ pA, $V_b = 20$ mV) on Ag(111). Value of $V_b$ for which $|\Delta f|$ is minimum corresponds to LCPD (see Methods). Grey curves: parabolic fitting of $\Delta f$ ($V_b$). Uniform fit residual indicates that tip-sample forces are mainly due to electrostatic interactions[27]. Inset: STM image of the coordination node in MOC with CO-functionalised tip ($I_t = 25$ pA, $V_b = 20$ mV). **b** LCPD map of tpy–$Fe_3$–tpy node relative to Ag(111), indicating negative charge accumulation at the centre of the node. **c** DFT-simulated electrostatic potential at the node, 3 Å above the molecular plane. **d** DFT-calculated energetically favourable geometry of tri-iron node (lateral view). **e** LCPD measured across tri-iron node [red; see dashed line in inset in **a**] and singly metalated tpy (green; dashed line in CO-tip STM image of Fe-TPPT-Fe in inset). Data recorded at constant height of 3 Å above that defined by STM set point ($I_t = 25$ pA, $V_b = 20$ mV) on Ag(111). Error bars: twice the s.d. **f** DFT-calculated charge density difference at the coordination node (top: vertical cut; bottom: horizontal; dashed horizontal lines indicate cut positions), show electron accumulation (vertical red arrows) between cations, ligand and surface. White circles indicate positions of N atoms of left tpy of the node. Dashed vertical lines indicate Fe atom positions. Frequency shift and LCPD measurements performed with CO-functionalised Pt/Ir tip. Scale bars: 2 Å

atom in a metal-organic system can be ambiguous in the framework of quantum mechanics[29,30], we gained further insight into the chemical environment of each Fe atom via complementary X-ray absorption measurements (Supplementary Fig. 11). While the node is electrostatically negative, we deduced from these measurements that the chemical environment of the distal Fe resembles that of Fe(II) in a 2+oxidation state, whereas for central Fe, it is similar to that of Fe(0) or Fe(I). This is consistent with DFT-based Bader analysis (see Supplementary Fig. 11 and related discussion in SI), which yields a positive charge state for all three Fe atoms, with the distal Fe (Bader charge ~ +0.8e) differing significantly from the central (~ +0.15e). The cluster has mixed valence, and the chemical state and reactivity of the cations—central Fe less positive than distal—depends on their location in the trimer (but not on their adsorption site on the surface; see SI). The negative charge accumulation at the node results from overscreening of the cations, mainly by the Fe–Fe bond electrons, with some contribution from the surface due to central Fe bridging the conduction bath. These charge distribution properties and electrostatics would differ drastically if the node contained two Fe's instead of three (Supplementary Fig. 10).

## Discussion

In conclusion, we presented the bottom-up synthesis of a tri-iron metal-organic coordination complex on a metal surface. Importantly, our experiments—supported by DFT and ncAFM imaging simulation—directly demonstrate the mixed (positive-)valence tri-nuclear configuration of the coordination motif, as well as the local electron accumulation at the ligand-cation-surface interface. Our on-surface approach is crucial for the formation of this unusual tri-nuclear complex. The noble metal surface offers balanced molecule-surface interactions, confining the molecular and atomic precursors to 2D and enabling efficient adsorbate diffusion. This results in a planar coordination motif, dramatically different from 3D, single-cation complexes in solution[23]. The coordination is stabilised by the site-specific charge redistribution at the metal-organic node.

This mixed-valence configuration can lead to specific on-surface sites that potentially exhibit local catalytic activity[31,32]. The self-assembly protocol can further allow for the design of well-defined 2D arrays of active multinuclear sites[32] with nanoscale precision, providing an attractive platform for solid-state catalysis. Indeed, catalytic function based on transition metal complexes relies on reduction–oxidation (redox) reactions between well-defined, ligand-anchored metal centres and reactants. These reactions benefit from the multiple spin and oxidation states (and the facile and reversible variations of the latter) of d-block metals[33]. Transition metal catalysis is paramount in biology and chemical applications, with polynuclear complexes offering useful and versatile reactivity patterns. For example, poly-iron compounds allow for oxygen fixation and C–H bond hydroxylation[34,35]. Di-ruthenium compounds catalyse the amination of C–H bonds[9,36]. In particular, compounds hosting metal–metal bonds, such as the MOC here, can stabilise electronic configurations (e.g., low-valent states) dramatically different from those in mononuclear compounds, with cooperativity between metal atoms potentially enhancing reactivity and catalytic performance[33].

A direct experimental demonstration of the catalytic activity of our tri-iron coordination complex is beyond the scope of this study and requires further experiments. However, this metal-organic system exhibits properties (namely, first-row tri-nuclear transition metal centre with direct metal–metal bonds and mixed-positive valence) that are commonly sought after for catalytic functionality, as it is well-established by extensive previous work in the field of transition metal catalysis[33]. Furthermore, the quasi-

flat on-surface morphology of the system can facilitate interactions between the metal centre and potential reactants, enhancing catalytic function in comparison to three-dimensional analogues (where the metal centre is caged by ligands).

Further applications could also benefit from the magnetic properties of metal–metal interactions in these nodes[6,37].

## Methods

**Sample preparation.** The 1D metal-organic nanostructures were synthesised in ultrahigh vacuum (UHV) by sequential deposition of TPPT molecules (first) and iron atoms (second) from the gas-phase onto a clean Ag(111) surface (Mateck GmbH; prepared in UHV by repeated cycles of $Ar^+$ sputtering and annealing at 720 K). See Fig. 1a for schematic. Terpyridine–phenyl–phenyl–terpyridine (TPPT; HetCat Switzerland) molecules were sublimed at 550 K onto a clean Ag(111) substrate held at room temperature (RT), resulting in a deposition rate of $\sim 4 \times 10^{-4}$ molecules $nm^{-2} s^{-1}$. Iron atoms were subsequently deposited from the gas-phase with the substrate held at RT to allow for metalation of TPPT molecules and the self-assembly of Fe–TPPT chains. The ratio of pristine to metalated molecules was controlled by varying the TPPT-to-Fe stoichiometry. The quantity and length of the chains can be controlled by the overall amount of metalated molecules on the surface as well as by the annealing time at RT (the longer the RT annealing time is, the longer the metal-organic chains). Both molecular and atomic adsorbates were deposited at sub-monolayer coverages. STM lateral manipulation experiments (Fig. 3) required further deposition of Fe with the substrate at 7 K to allow for single Fe adatoms on the surface. The base pressure was below $2 \times 10^{-9}$ mbar during molecular deposition, and below $1 \times 10^{-10}$ mbar during Fe evaporation as well as for all characterisation measurements.

**Carbon monoxide tip functionalisation.** Carbon monoxide (CO) molecules were used to functionalise both the STM and ncAFM Pt/Ir tips as follows[38,39]. CO was dosed in situ with the sample maintained below 8 K (see Supplementary Fig. 1). By exposing the sample to a partial pressure $p(CO) = 5 \times 10^{-8}$ mbar, we reproducibly deposited CO at a rate of $\sim 0.001$ molecules $nm^{-2} s^{-1}$. Typical deposition times were 3 seconds. The tip was first positioned on top of a CO molecule at a height defined by the STM set point ($I_t = 25$ pA, $V_b = 20$ mV) and then brought 400 pm closer to the surface to pick up the molecule. Once the CO was transferred from the surface to the tip, we confirmed that the molecule was adsorbed symmetrically by imaging another CO on the surface as a circular protrusion surrounded by a symmetric depression[40].

**STM/STS measurements.** All STM and STS measurements were performed at 4.6 K with a Ag-terminated (blue colour mapping; Fig. 3) or a CO-functionalised Pt/Ir tip for increased resolution (brown colour mapping; Figs. 1, 2a–c, insets Fig. 4). All topographic images were acquired in constant-current mode. STS measurements were obtained by measuring the tunnelling current as a function of the tip-sample bias, scanning the region of interest pixel-by-pixel, with the tip height stabilised according to an STM set point at each location. The normalised numerical derivative $(dI/dV)/(I/V)$ spectra were computed as an approximation of the local density of states[41]. The sample bias is reported throughout the text.

**ncAFM measurements.** All ncAFM experiments were performed using a qPlus tuning fork sensor[42] (Createc) in frequency modulation mode at 4.6 K. (resonance frequency $f \sim 29$ kHz; spring constant $k \sim 1800$ N $m^{-1} \pm 7\%$), with a CO-terminated Pt/Ir tip. All ncAFM topography images were taken using a 0.6 Å amplitude at constant height. Various tip-sample distances were used, which were defined with respect to an STM set point on Ag(111) [$I_t = 25$ pA, $V_b = 20$ mV]. No bias voltage was applied during the ncAFM topographic measurements. All ncAFM topographic images were filtered by Laplace edge detection; see Supplementary Figs. 4 and 5 for details.

**LCPD measurements.** LCPD measurements were performed with a qPlus tuning fork sensor (Createc) with an oscillation amplitude of 1 Å. All data were recorded at a constant height with respect to an STM set point on Ag(111). The LCPD $V^*$ (defined as the sample bias voltage $V_b$ for which the absolute value $|\Delta f(V_b = V^*)|$ of the resonance frequency shift is minimum) was determined by measuring $\Delta f$ as a function of $V_b$ (applied to the sample). The LCPD is plotted relative to that of the Ag substrate. In order to be sensitive to tip-sample forces that are dominantly electrostatic in nature, we performed LCPD measurements at significantly large tip-sample distances, for which $\Delta f(V_b)$ curves could be fitted with a parabola[43], with a minimal random noise residual[27] (Fig. 4a). In these conditions, the LCPD $V^*$ is a measure of the variations of the electrostatic potential—and hence of the variations in charge distribution—at the surface. If we consider two different locations $r_A$ and $r_B$ on the surface, and $V^*(r_A)$ and $V^*(r_B)$ are positive (with respect to a zero-reference potential, in this case Ag), then there is an accumulation of negative charge at $r_A$ and $r_B$, and if $|V^*(r_A)| > |V^*(r_B)|$, then that charge accumulation is larger at $r_A$ than at $r_B$. The Pt/Ir tip was functionalised with a CO molecule for all ncAFM/LCPD measurements.

**STM lateral manipulation.** All STM lateral manipulation[44] experiments (Fig. 3) were performed at 4.6 K with an Ag-terminated Pt/Ir tip. The tip was approached onto a bare patch of Ag(111) by changing the STM set point ($I_t = 30$ pA, $V_b = -10$ mV for displacing Fe adatoms; $I_t = 100$ nA, $V_b = -10$ mV for breaking the coordination node) and subsequently moved laterally at a rate of $\sim 500$ Å $s^{-1}$.

**DFT calculations.** All DFT calculations were performed using the Vienna Ab initio Simulation Package (VASP)[45,46], with the projector augmented plane-wave method[47] and PBE exchange correlation functional. The empirical Grimme D2[48] correction with default parameters was used to model van der Waals interactions. The energy relaxation of the tri-iron metal-organic node was performed using[49] L(S)DA + U with two different values for $U$ (3.0 and 5.0 eV) to check the robustness of the results. Both calculations yielded the same symmetric node geometry, with Fe–Fe bond lengths of $\sim 2.4$ Å (in agreement with experiments) and anti-ferromagnetic ordering of Fe magnetic moments. It is important to note that calculations that did not include magnetic interactions or that included a ferro-magnetic configuration led to asymmetric node geometries with significantly shorter Fe–Fe bond lengths, in disagreement with our experiments. The simulation supercell consisted of a rectangular Ag(111) slab of $17 \times 8$ atoms, containing two units of the molecular chain (total size: $49 \times 20 \times 20$ Å$^3$). Initial geometric relaxations were performed with one fixed layer of Ag atoms. Simulations of ncAFM images (Fig. 2), electronic potential and charge density difference (Fig. 4) were obtained by considering three Ag layers and relaxing the electronic degrees of freedom while keeping the atom positions fixed. Charge densities and electrostatic potential did not exhibit any notable changes after adding Ag layers. The charge density difference corresponds to the subtraction between the charge density of the interacting system and that of the system composed of non-interacting atoms in the same positions as the relaxed system. Bader charge analysis was based on Ref.[50].

**ncAFM imaging simulations.** Simulated ncAFM images were obtained using the ProbeParticle code[28,51] with default van der Waals parameters (see Supplementary Table 2). The electrostatic potential was modelled as the electrostatic interaction between the DFT-relaxed system (see above) and the quadrupole model density[52] on the probe particle (quadrupole strength $Q = -0.4$; stiffness of CO molecule: 0.5 N/m). Discrepancies between simulated and experimental images can be explained by: (i) overestimation of the delocalisation of $\pi$ electrons by DFT, resulting in flatter aromatic systems;[53] (ii) omission of long-range background forces in simulations, leading to a low-frequency shift on the substrate; and (iii) underestimation of the interactions between free electron pairs of N atoms of pristine TPPT with the metal surface, leading to bright protrusions in Fig. 2g.

**Data availability.** The data supporting the findings of this study are available from the authors upon reasonable request.

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

## Acknowledgements

We thank Felix Bischoff for contributing to the X-ray absorption measurements and acknowledge support from the German Academic Exchange Service (DAAD). M. Castelli, D.K. and A.S. acknowledge support from the Monash Centre of Atomically Thin Materials (MCATM). D.K. acknowledges support from the Australian Research Council (ARC) Centre of Excellence in Future Low-Energy Electronics Technologies. A.S. acknowledges support from the ARC Future Fellowship scheme (FT150100426). M. Capsoni and S.A.B. acknowledge support from the ACS Petroleum Research Fund (55955-ND5). P.J. acknowledges support from Praemium Academie of the Czech Academy of Sciences and the Ministry of Education of the Czech Republic Grant LM2015087 and GACR project No. 18-09914S. P. H. acknowledges support from the Czech Academy of Sciences project MSM100101705.

## Author contributions

C.K., M.C., D.K. and A.S. designed and performed all experiments, and analysed and interpreted all data. P.H. and P.J. performed the theoretical modelling and calculations. A.T., M.C., M.E., J.H. and S.A.B. contributed to the X-ray absorption measurements. C. K., M.C., and A.S. wrote the manuscript. All authors discussed the results and contributed to the final manuscript.

## Additional information

**Competing interests:** The authors declare no competing interests.

