## [Peer Review File · Nature Communications]

Reviewers' comments:

Reviewer #1 (Remarks to the Author):

Krull et al report a work using STM/AFM to characterize the metal-organic complexes assembled on a surface. They applied high-resolution AFM and DFT calculations to determine that there three co-linearly arranged Fe atoms in a complex. Furthermore, they used STM tip to manipulate the complexes, which confirms the three-Fe scheme. In this regard, this work provides clear evidences for a multi-nuclear coordination example. Another new result is the DFT calculated Bader charge of the distal/central Fe is +0.8e/+0.15e. However, besides these, the novelty of the work is not outstanding. As mentioned by the authors, the multi-nuclear coordination systems have been reported before. The major insight claimed by the authors is the mixed valence configuration of the three Fe atoms may exhibit local catalytic activity. This is speculative but not supported by any data. Overall, the reviewer does not think the manuscript meets the criteria of Nature Communication.

Reviewer #2 (Remarks to the Author):

It is an interesting paper, which introduces the on-surface synthesis of 1D metal-organic nanostructures composed of terpyridine (tpy)-based molecules and coordinated tri-iron clusters. Using a combined approach of STM, nc-AFM and DFT calculations, the authors suggested the formation of linear iron linkages is due to the local accumulation of electrons at the interface between iron cations, molecular ligand and metallic substrate. In my viewpoint, this is a well-done experiment with persuasive data. I would recommend the paper for publication should the authors fully addressed the following issues.

1. Apparently the tri-iron cluster is not in a linear configuration. Wasn't it a V shape as illustrated in Fig.4d?
2. A rigorous explanation should be given for the mixed valence configuration of the tri-iron cluster, in view of the discussions on catalytic activity and magnetic properties.
3. Why dI/dV mappings for Fig.3f-i were not presented? It would be more straightforward for explaining the electronic states at the tri-iron coordination node.
4. line 98, it claims that "the pyridine tilted towards the surface due to interactions between the nitrogen lone electron pairs and noble metal" and cited Ref.19. But the DFT calculations in Ref.19 found TPPT in a near planar adsorption on Ag and only a small twist between two phenyls.
5. Fig.S5 showed new peaks in the valleys after Laplace-filtering, but the peaks were not seen in raw data. Are these filter-generated peaks convincing?
6. Based on Ref.23, Au adatom in the metal-organic complex appeared as bright protrusions in STM images. Why was the Fe adatom in tpy-Fe complex featureless in STM image (line 109)?
7. "the nc-AFM feature of the central Fe was originated from the electrostatic field rather than from topography". This is partially inconsistent with the experimental result. The contrast of central Fe fade away at $Z = 0.4 \text{ \AA}$, and cannot be identified in AFM image.
8. line 123, Ref.25 was about a reaction on Au(111), not on Cu(111).

Reviewer #3 (Remarks to the Author):

The manuscript entitled "Iron-based trinuclear metal-organic nanostructures on a surface with local charge accumulation" describe the one-dimensional self-assembly on Ag(111) substrate of a tpy-based aromatic molecule (tpy-phenyl-phenyl-tpy; TPPT) coordinated with Fe adatoms. The authors combine the use of low-temperature scanning tunnelling microscopy (STM), spectroscopy (STS), non-contact atomic-force-microscopy (ncAFM), local contact-potential-difference (LCPD) measurements, density-functional-theory (DFT) and ncAFM image simulations to elucidate the atomic scale morphology and local charge distribution of the coordination node. They directly demonstrate that the metal-ligand coordination motif consists of flat, coplanar tpy's linked via a linear tri-iron cluster and that the Fe atoms in this coordination node are in a cationic mixed valence configuration, exhibiting direct metal bonding including a the charge redistribution with the electrons accumulated at the cation-ligand-surface interface.

The manuscript is very well write and easy to read and follow and the author present the results in a very clear manner, in addition there is an extended and very detailed supplementary information containing very important experimental hints and explanation of data analysis. They use not only the appropriate techniques but also novel ones (as CO tip modification for both STM and AFM) and with carefully data analysis as LPCD.

I recommend the article for publication in Nature Communication. However a major concern from my side should be address in order to improve the quality of the manuscript.

The authors do not discuss in detail the role of the substrate. Not only on the molecule assembly (because I was discussed in ref 19). In order to discuss better the local charge redistribution at the metal organic node. Does the 1D self-assembly follow the substrate geometry? Could the authors tell something about in which position (top/ hollow/ bridge) are the Fe atoms sitting? Could be also this a reason of diferent individual oxidation state for the three metal centers? This information could be achieve by the DFT data. Is any other evidence about oxidation state?

We thank all three referees for their constructive insights and comments, which have allowed us to improve our manuscript substantially. Below we have addressed general remarks and point-by-point comments from each referee.

Reviewer #1:

Krull et al. report a work using STM/AFM to characterize the metal-organic complexes assembled on a surface. They applied high-resolution AFM and DFT calculations to determine that there three co-linearly arranged Fe atoms in a complex. Furthermore, they used STM tip to manipulate the complexes, which confirms the three-Fe scheme. In this regard, this work provides clear evidences for a multi-nuclear coordination example. Another new result is the DFT calculated Bader charge of the distal/central Fe is +0.8e/+0.15e. However, besides these, the novelty of the work is not outstanding. As mentioned by the authors, the multi-nuclear coordination systems have been reported before. The major insight claimed by the authors is the mixed valence configuration of the three Fe atoms may exhibit local catalytic activity. This is speculative but not supported by any data. Overall, the reviewer does not think the manuscript meets the criteria of Nature Communication.

Answer to Reviewer #1:

Reviewer #1 claims that the ‘novelty of the work is not outstanding’, and that ‘multinuclear coordination systems have been reported before’. We hope that in the following we are able to convince Reviewer #1 of the novel aspects and quality of our work, as well as its relevance for the broad readership of Nature Communications, as pointed out by Reviewers #2 and #3. We acknowledge that multinuclear metal-organic compounds have already been synthesized and characterized in previous work. Indeed, research in multinuclear complexes with metal-metal bonds was pioneered by work from F. A. Cotton et al. started in the 1960’s; it represents a field of research in itself. This is highlighted in our manuscript and cited references. It is an important area of research in inorganic chemistry due to the vast range of applications in metal-organic frameworks, photosensitizers, magnetism, molecular-based nanoelectronics, catalysis, etc. (e.g., see Ref. 8 in our manuscript).

Reviewer #1 states that “the major insight claimed by the authors is that the mixed valence configuration of the three Fe atoms may exhibit local catalytic activity”. The novelty of our work and its importance resides in two other key aspects. First, we have demonstrated the possibility to synthesize multinuclear coordination complexes from the bottom-up using protocols of on-surface supramolecular chemistry, with configurations and properties that would not be feasible with more conventional synthetic chemistry approaches without a surface. We have added a sentence at the end of the first paragraph of page 4 to emphasize this aspect. Second, we have shown that this strategy allows for a full and direct characterization of the atomic-scale structural and electrostatic properties of this novel compound, via a combination of low-temperature scanning probe microscopy techniques and first-principles calculations. It is important to note that it is the first time that this combination of techniques is used for a precise, atomic-level characterization of a multinuclear, metal-metal coordination compound.

Reviewer #1 further claims that ‘local catalytic activity ... is speculative but not supported by any data’. For more than 50 years now, there has been a great interest in the synthesis of multinuclear coordination complexes. As mentioned above, this interest has been motivated by the vast range of applications that such compounds allow for, in particular catalysis. It is well established that catalytic functionality can benefit from and can be enhanced by cooperativity of metal-metal polynuclear compounds. Examples of this include dehydrogenation of primary alcohols with di-ruthenium compounds,¹ activation of methane by di-tantalum complexes,² production of dihydrogen as sustainable alternative fuel by di-rhodium compounds,³ photoelimination of nitrides by di-ruthenium complexes,⁴ etc. (see further examples in the references cited in the main text). Despite the relevance for such applications of polynuclear complexes based on well-defined transition metal clusters (in particular Group VIII metals), no quasi-linear metal-metal tri-iron compound has been reported. Importantly, no atomic-scale characterization (structural and electronic) of such compounds has been achieved. Our work bridges this gap.

We acknowledge that our work does not provide a direct experimental demonstration of catalytic activity for the reported compound. However, it is important to note that the chemical reactivity and resulting catalytic activity of these compounds is determined by their exact, atomic-scale morphology, electronic configuration and electrostatic environment. Our work not only demonstrates the synthesis of a novel trinuclear coordination compound, but it also fully describes its atom-scale properties. Our combination of ncAFM measurements, LCPD mapping, DFT calculations and X-ray absorption data [see Supplementary Figure 11 added to Supplementary Information (SI)] reveal the complex charge distribution at the Fe trimer, supporting the hypothesis of enhanced chemical reactivity. A direct experimental demonstration of the chemical reactivity and catalytic activity of this compound is beyond the scope of this study. Our study provides the grounds and calls for further experiments for such direct demonstration. We added a few sentences at the end of the main text on this matter.

Reviewer #2:

It is an interesting paper, which introduces the on-surface synthesis of 1D metal-organic nanostructures composed of terpyridine (tpy)-based molecules and coordinated tri-iron clusters. Using a combined approach of STM, ncAFM and DFT calculations, the authors suggested the formation of linear iron linkages is due to the local accumulation of electrons at the interface between iron cations, molecular ligand and metallic substrate. In my viewpoint, this is a well done experiment with persuasive data. I would recommend the paper for publication should the authors fully address the following issues.

We thank Reviewer #2 for her/his positive feedback. We address her/his concerns below.

1. Apparently the tri-iron cluster is not in a linear configuration. Wasn't it a V shape as illustrated in Fig.4d?

We agree with Reviewer #2 that the tri-iron node is not perfectly linear, but slightly kinked, with the central iron atom closer to the surface. We thank the Reviewer for this comment. We have

specified this in the manuscript by replacing ‘linear’ with ‘quasi-linear’ throughout the text, replacing ‘linearly arranged Fe atoms’ with ‘Fe atoms arranged quasi-linearly in a plane perpendicular to the surface’ in the 1st paragraph of page 7, and adding ‘, resulting in a quasi-linear, slightly kinked tri-iron cluster’ on page 8.

2. A rigorous explanation should be given for the mixed valence configuration of the tri-iron cluster, in view of the discussions on catalytic activity and magnetic properties.

We agree with Reviewer #2 that the explanation of the mixed valence configuration of the tri-iron node might require more detail. We thank Reviewer #2 for her/his comment. To address this, we performed complementary near-edge X-ray absorption fine structure (NEXAFS) spectroscopy measurements to gain further insight into the effective chemical environment of the Fe atoms in the trinuclear coordination motif. We added details on these measurements as well as related results and discussion in the SI. We also added two sentences related to these measurements in the 1st paragraph of page 9 of the main text. Although assigning an integer oxidation state to a metal atom in a metal-organic system can be ambiguous in the framework of quantum mechanics (see references added to main text and SI), these measurements show, in combination with the DFT-based Bader analysis, that: (i) the distal Fe atoms in the tri-iron node (which coordinate and directly interact with the TPPT molecules) are charged positively and have a chemical environment similar to that of Fe(II) in a 2+ oxidation state; (ii) the chemical environment of the central Fe is similar to that of Fe(0) or Fe(I) in a neutral or 1+ oxidation state. This corroborates the mixed positive valence of the tri-iron node.

3. Why dI/dV mappings for Fig.3f-i were not presented? It would be more straightforward for explaining the electronic states at the tri-iron coordination node.

The main purpose of Figs. 3f-i is to demonstrate that the tpy-Fe^(A) (tpy-Fe₂^(A)) and tpy-Fe^(D) (tpy-Fe₂^(D), respectively) complexes are not only morphologically identical, but also electronically, with the same electronic fingerprint. This is shown clearly and unambiguously by our dI/dV (V_b , x) plots as a function of bias voltage V_b and position x along the molecular axis, with one plot for each complex. For this purpose, 2D dI/dV (x , y) maps at different bias voltages V_b are not strictly required. Moreover, the latter would require much more manuscript space, with several panels for each complex. It is important to note that a detailed discussion on dI/dV data on this system is beyond the scope of this manuscript and will be addressed in a subsequent publication that we are currently preparing.

4. line 98, it claims that “the pyridine tilted towards the surface due to interactions between the nitrogen lone electron pairs and noble metal” and cited Ref.19. But the DFT calculations in Ref. 19 found TPPT in a near planar adsorption on Ag and only a small twist between two phenyls.

Our ncAFM measurements and DFT calculations show a very slight out-of-plane tilt of the pyridine groups of the pristine TPPT towards the Ag(111) surface ($\sim 3^\circ$ and $\sim 1.5^\circ$, respectively; see Supplementary Table 1 in SI). The results of our DFT calculations here are hence consistent with a quasi-flat geometry of the terpyridine group of pristine TPPT. It is important to note that the STM measurements of TPPT/Ag(111) in Ref. 19 do not allow to resolve this very subtle intramolecular tilt. The interaction between the pyridine nitrogen lone electron pairs and noble

metal surface, which mediates the very subtle pyridine tilt, is explicitly mentioned in Ref. 19 (see Fig. 3 in Ref. 19). Our results here on TPPT/Ag(111) are hence consistent with those of Ref. 19. To emphasize that this out-of-plane tilt is very subtle, we replaced “tilted” with “very subtly tilted” in the 2nd paragraph of page 5, and “tilted” with “slightly tilted” in the 2nd paragraph of page 7 of the main text.

5. Fig.S5 showed new peaks in the valleys after Laplace-filtering, but the peaks were not seen in raw data. Are these filter-generated peaks convincing?

By definition, Laplace-filtering of a 2D image [function $f(x, y)$] corresponds to the square of the gradient of image $f(x, y)$; $\Delta f(x, y) = \nabla^2 f(x, y) = \frac{\partial^2 f(x,y)}{\partial x^2} + \frac{\partial^2 f(x,y)}{\partial y^2}$. Therefore, a Laplace-filtered image emphasizes edges and peaks, resulting in an enhancement of contrast. That is, the relative height of peaks in a Laplace-filtered image becomes larger than that in the original image. However, it is important to note that the position of peaks and features in the original image is not altered by Laplace-filtering. For example, the ncAFM height profile across the tri-iron node along the molecular axis in Supplementary Figure 5b (blue curve) shows five prominent peaks. The height of these peaks becomes larger by Laplace-filtering, although the peak positions are not altered (blue curve in Supplementary Figure 5d). Because of the contrast enhancement, weaker features present in the raw data can become more apparent by Laplace-filtering. We have updated figure S5 and added an additional sentence to the SI to highlight this.

6. Based on Ref. 23, Au adatom in the metal-organic complex appeared as bright protrusions in STM images. Why was the Fe adatom in tpy-Fe complex featureless in STM image (line 109)?

Line 109 of our manuscript is related to ncAFM imaging of the tpy-Fe complex, not STM.

The Fe adatom in the tpy-Fe complex is not featureless in all of our metal-tip STM, CO-tip STM and CO-tip ncAFM measurements. This becomes clear when comparing imaging of the pristine tpy group with imaging of the metalated tpy-Fe group. For example, metal-tip STM imaging of tpy-Fe at $V_b = -500$ mV (Fig. 3c) shows a “hammer” shape, with a bright protrusion at the centre, very different from the “v-shaped” pristine tpy (see Ref. 19 of main text by Capsoni et al.). This difference is also clearly visible in CO-tip STM imaging; compare “hammer”-shaped tpy-Fe in Fig. 2b with pristine “v-shaped” tpy in Figs. 2b and 2a. This is consistent with Ref. 23 from Pawlak et al. (Fig. 1g therein) where STM imaging of the tpy-Au complex shows a clear bright protrusion at the centre of the tpy group. Note that STM imaging of tpy-Fe is bias dependent; Fe-related features at the centre of tpy become more apparent at negative sample bias (e.g., when $V_b < -100$ mV; see Fig. 3c in our manuscript), whereas at positive bias tpy-Fe becomes more similar to the “v-shaped” pristine tpy. CO-tip ncAFM imaging also shows a clear difference between pristine tpy and the tpy-Fe complex. See apparent height profiles in Fig. 2e, where pristine tpy shows a protrusion which is absent for the darker tpy-Fe. This is also consistent with Ref. 23 from Pawlak et al. (see Fig. 3 therein).

7. “the ncAFM feature of the central Fe was originated from the electrostatic field rather than from topography”. This is partially inconsistent with the experimental result. The contrast of central Fe fade away at $Z = 0.4$ Å, and cannot be identified in AFM image.

Indeed, as the tip-sample distance z decreases and reaches 0.4 \AA , the ncAFM imaging contrast of the central Fe vanishes (Fig. S3). This is fully consistent with our claim that the protrusion seen in ncAFM imaging at the centre of the tri-iron linkage (Fig. 2f) is due to electrostatic interactions rather than topography. First, this central protrusion can be reproduced accurately by our ncAFM simulations if the electrostatic interactions are included in the simulations of ncAFM images. It cannot be reproduced accurately if the CO molecule is uncharged (see Supplementary Figure 9), i.e., if tip-sample electrostatic interactions are not taken into account. Second and most importantly, these ncAFM simulations including electrostatic tip-sample interactions fully reproduce our z -dependent experimental ncAFM data of the node (Supplementary Figure 7), in particular for smaller z values where the central protrusion vanishes. Indeed, Supplementary Fig. 7 (b: experiment; c-g: simulations) shows that, while for significantly “large” values of z (e.g., $z \sim 0 \text{ \AA}$ in Supplementary Fig. 7) the frequency shift Δf at the central Fe is larger than at the surrounding ligands (resulting in the central protrusion in ncAFM imaging), the contrast is reversed when z decreases (e.g., $-1 < z < -0.8 \text{ \AA}$), with Δf becoming smaller at the centre Fe than at the organic moiety, resulting in the vanishing of the protrusion seen in Supplementary Fig. 3(l). This z -dependent variation of the contrast is fully reproduced in our ncAFM simulations including electrostatic forces between tip and sample, where the CO-tip is described with a quadrupole charge model [for details see J. Peng et al. Nature Communications 9, 122 (2018)]. To clarify this, we added simulated ncAFM images for different values of z in Supplementary Fig 7.

8. line 123, Ref.25 was about a reaction on Au(111), not on Cu(111).

We thank the referee for pointing out this error. We have corrected the manuscript accordingly by replacing Cu with Au: “This provides evidence that tpy-Fe₂^(D) consists of a tpy coordinated to two Fe adatoms, similar to di-nuclear poly-pyr coordination chains on Au(111).”

Reviewer #3:

The manuscript is very well write and easy to read and follow and the author present the results in a very clear manner, in addition there is an extended and very detailed supplementary information containing very important experimental hints and explanation of data analysis. They use not only the appropriate techniques but also novel ones (as CO tip modification for both STM and AFM) and with carefully data analysis as LPCD. I recommend the article for publication in Nature Communication.

We thank Reviewer #3 for her/his positive feedback. We address her/his concerns below.

However a major concern from my side should be address in order to improve the quality of the manuscript. The authors do not discuss in detail the role of the substrate. Not only on the molecule assembly (because I was discussed in ref 19). In order to discuss better the local charge redistribution at the metal organic node. Does the 1D self-assembly follow the substrate geometry? Could the authors tell something about in which position (top/ hollow/ bridge) are the Fe atoms sitting? Could be also this a reason of different individual oxidation state for the three

metal centers? This information could be achieved by the DFT data. Is any other evidence about oxidation state?

We agree with Reviewer #3 that the role of the substrate was not discussed in detail in the manuscript. To address this, we have: (i) added two sentences in the main text (see 2nd paragraph of p. 4 and 1st paragraph of p. 6) and a section in the SI showcasing that the metal-organic coordination and chain formation reduces the interaction between molecule and surface, evidenced by the apparent lack of well-defined orientations of the MOCs with respect to the substrate crystalline axes in comparison with the case of the pristine TPPT molecule; (ii) provided details on the relaxed adsorption geometry of the tri-iron coordination node in the MOCs based on DFT calculations, showing that each of the Fe atoms in the node is located close to a Ag(111) hollow site (see added sentence in 2nd paragraph of p. 7 of main text). It is important to note that the simulated ncAFM images and electrostatic potential derived from these calculations are consistent with our experimental data.

As with Reviewer #2, we further agree with Reviewer #3 that the chemical environment (that is, oxidation state) of each of the Fe atoms in the metal-organic coordination node might require a more detailed explanation. We thank Reviewer #3 for her/his question on this matter. To address this, we performed complementary near-edge x-ray absorption fine structure (NEXAS) spectroscopy measurements to gain insight into the effective chemical environment of the Fe atoms. We added details on these measurements as well as related results and discussion in the SI. We also added a few sentences related to these measurements in the 1st paragraph of page 9 of the main text. It is important to note that assigning an integer oxidation state to a metal atom in a metal-organic system can be ambiguous in the framework of quantum mechanics (see references added to main text and SI). These NEXAFS measurements show, in combination with the DFT-based Bader analysis, that: (i) the distal Fe atoms in the tri-iron node (which coordinate and directly interact with the TPPT molecules) are charged positively and have a chemical environment close to that of Fe(II) in a 2+ oxidation state; (ii) the chemical environment of the central Fe is similar to that of Fe(0) or Fe(I) in a neutral or 1+ oxidation state. This corroborates our claim of the mixed positive valence of the tri-iron node.

Our DFT calculations (which reproduce all of our experimental ncAFM and LCPD data) support a slightly kinked, v-shaped Fe trimer configuration, with the central Fe slightly closer to the surface in comparison to the two distal Fe (Fig. 4f of main text). Therefore, the interaction between the substrate and the distal Fe atoms is arguably weaker than that for the central Fe (if there is any difference at all). This, in combination with the fact that the adsorption site of each of the Fe atoms is quasi-identical [i.e., close to a Ag(111) hollow site; see above], provides a strong indication that the significant difference in chemical environment between central and distal Fe atoms cannot be explained by a difference in adsorption site. It is a result of the metal-organic coordination. We added a paragraph in the SI to emphasize this.

Regarding the Bader charge analysis for each of the Fe atoms in the trimer (central: $\sim +0.15e$; distal: $\sim +0.8e$), we should stress that the DFT-calculated electrostatic potential in Fig. 4c is the result of a rather complex non-spherical charge distribution around the Fe atoms. Therefore, the resulting point charge for each individual Fe atom may vary depending on the used projection (i.e., Bader, Mulliken, Löwdin). Our Bader analysis serves as a tentative approach to highlight

the differences in chemical environment and valence state between distal and central Fe atoms. It does not allow for an accurate and unambiguous determination of a point charge (i.e., oxidation state), which, in any case, would not be meaningful within the framework of quantum mechanics. This DFT-derived Bader analysis is consistent with our X-ray absorption measurements. We added a small paragraph in the SI discussing this matter.

1. Dutta I, Sarbajna A, Pandey P, Rahaman SMW, Singh K, Bera JK. Acceptorless Dehydrogenation of Alcohols on a Diruthenium(II,II) Platform. *Organometallics* **35**, 1505-1513 (2016).
2. Li HF, *et al.* Methane Activation by Tantalum Carbide Cluster Anions Ta₂C₄⁻. *J Phys Chem Lett* **8**, 605-610 (2017).
3. Heyduk AF, Nocera DG. Hydrogen produced from hydrohalic acid solutions by a two-electron mixed-valence photocatalyst. *Science* **293**, 1639-1641 (2001).
4. Corcos AR, Pap JS, Yang T, Berry JF. A Synthetic Oxygen Atom Transfer Photocycle from a Diruthenium Oxyanion Complex. *J Am Chem Soc* **138**, 10032-10040 (2016).

REVIEWERS' COMMENTS:

Reviewer #2 (Remarks to the Author):

I'm ok with the authors' reply and the revised manuscript.

Reviewer #3 (Remarks to the Author):

The manuscript entitled "Iron-based trinuclear metal-organic nanostructures on a surface with local charge accumulation" describe the one-dimensional self-assembly on Ag(111) substrate of a tpy-based aromatic molecule (tpy-phenyl-phenyl-tpy; TPPT) coordinated with Fe adatoms. The authors combine the use of low-temperature scanning tunnelling microscopy (STM), spectroscopy (STS), non-contact atomic-force-microscopy (ncAFM), local contact-potential-difference (LCPD) measurements, density-functional-theory (DFT) and ncAFM image simulations to elucidate the atomic scale morphology and local charge distribution of the coordination node. They directly demonstrate that the metal-ligand coordination motif consists of flat, coplanar tpy's linked via a linear tri-iron cluster and that the Fe atoms in this coordination node are in a cationic mixed valence configuration, exhibiting direct metal bonding including a the charge redistribution with the electrons accumulated at the cation-ligand-surface interface.

In the first round I was already recommended the manuscript for publication with minor revisions. However, the authors did a huge improvement of the manuscript by following carefully all reviews comments. In my personal case, they added same sentences in the manuscript to make it more clear and even they added references to reinforce their conclusions.

I fully recommend the article for publication in Nature Communication.